# The SpasT-SCI-T trial protocol: Investigating calpain-mediated sodium channel fragments as biomarkers for traumatic CNS injuries and spasticity prediction

Guillaume Baucher[1,2]*, Sylvie Liabeuf[1◉], Cécile Brocard[1◉], Aurélie Ponz[3], Karine Baumstarck[3], Lucas Troude[2], Marc Leone[4], Pierre-Hugues Roche[1,2], Frédéric Brocard[1]*

1 Institut de Neurosciences de la Timone, Aix-Marseille Université, CNRS, Marseille, France, 2 AP-HM, Hôpital Universitaire Nord, Neurochirurgie adulte, Marseille, France, 3 AP-HM, Direction de la Recherche Santé et Maladies Rares, Marseille, France, 4 AP-HM, Hôpital Universitaire Nord, Service d'Anesthésie et de Réanimation, Marseille, France

◉ These authors contributed equally to this work.
* guillaume.baucher@ap-hm.fr (GB); frederic.brocard@univ-amu.fr (FB)

## Abstract

Spinal cord injury and traumatic brain injury are major causes of long-term disability and are often complicated by spasticity, a motor disorder characterized by increased muscle tone and exaggerated reflexes that significantly impair quality of life. Current diagnostic methods lack the sensitivity needed to accurately predict the severity of injury or the onset and progression of spasticity. Trauma-induced calcium dysregulation activates calpains, a family of proteases that cleave sodium channels, disrupting their inactivation and increasing persistent sodium currents. This cascade drives the overexcitability of motoneurons, contributing to the development of spasticity. Consequently, sodium channel fragments have emerged as promising biomarkers that link injury mechanisms to clinical outcomes. The present SpasT-SCI-T clinical trial protocol aims to evaluate sodium channel fragments as blood biomarkers for assessing the severity of spinal cord and traumatic brain injuries, as well as their potential to predict clinical outcomes, including the development of spasticity. This prospective, multicenter, case-control and cohort study involves 40 participants: 20 individuals with spinal cord injury, 10 individuals with traumatic brain injury, and 10 healthy controls. Blood samples are collected within six hours of injury and at follow-up points over six months. Clinical outcomes, including spasticity (assessed using the Modified Ashworth Scale), neurological recovery (measured by the American Spinal Injury Association Impairment Scale and Glasgow Coma Scale), and quality of life (evaluated using the Short Form-36 Health Survey), are analyzed in correlation with biomarker levels. We anticipate that calpain-mediated sodium channel fragments will transform the management of central nervous system injuries by enabling early diagnosis, improving prognostic accuracy, and guiding personalized therapeutic

**Data availability statement:** No datasets were generated or analysed during the current study. All relevant data from this study will be made available upon study completion.

**Funding:** This study was supported by the French National Research Agency (ANR) under grant number ANR-21-CE17-0060, awarded to FB. This public research funding body did not play any role in the study design, data collection and analysis, decision to publish, or preparation of the manuscript.

**Competing interests:** The authors have declared that no competing interests exist.

strategies. The clinical trial is registered on ClinicalTrials.gov (NCT06532760, January 10, 2024), with Assistance Publique–Hôpitaux de Marseille as the sponsor.

## Introduction

Spinal cord injury (SCI) is a tragic condition with lifelong physical, psychological, and economic consequences, affecting millions of individuals worldwide [1–3]. Accurate and timely assessment of its severity is paramount for predicting outcomes and guiding therapeutic interventions. However, current diagnostic methods, which rely primarily on clinical examinations and imaging, lack the sensitivity required for precise evaluation, especially during the critical acute phase [4–6]. This diagnostic gap underscores the urgent need for novel, reliable biomarkers to improve clinical decision-making.

One of the most prevalent and disabling complications of SCI is spasticity, which affects up to 75% of patients within the first year post-injury [7,8]. Spasticity, classically defined as hypertonia with a velocity-dependent increase in stretch reflexes [9], encompasses a spectrum of clinical manifestations, including clonus, exaggerated reflexes, muscle spasms, and involuntary leg crossing [10,11]. These symptoms severely disrupt daily activities, quality of life, and rehabilitation efforts. Without early and effective management, spasticity progresses to complications such as muscle shortening, joint contractures, and painful pressure sores [12,13]. The unpredictable onset and variable severity of spasticity, ranging from mild stiffness to severe, uncontrollable spasms, pose significant challenges for clinicians, emphasizing the urgent need for reliable predictive biomarkers [14,15].

Advances in our understanding of spasticity pathophysiology have highlighted promising targets for biomarker discovery [16,17]. Secondary injury processes in SCI trigger molecular cascades, including excessive glutamate release, intracellular calcium overload, and calpain activation [18,19]. Calpains, calcium-activated proteases, cleave Nav1.6 voltage-gated sodium channels [20–22], which are predominantly expressed in spinal motoneurons [23–25]. This proteolysis disrupts the normal inactivation of Nav1.6 channels, leading to an upregulation of persistent sodium currents ($I_{NaP}$) [20–22]. The pathological $I_{NaP}$ drives motoneuron hyperexcitability and sustained depolarizations (plateau potentials), directly contributing to spasticity in both animals [26–31] and humans [32–34].

The identification of sodium channel fragments as products of calpain-mediated cleavage offers a novel biomarker opportunity, linking the molecular pathology of SCI to clinical outcomes. These fragments hold potential not only for assessing SCI severity but also for predicting the onset and magnitude of spasticity [21]. Notably, similar sodium channel cleavage mechanisms have been observed in traumatic brain injury (TBI), suggesting broader applicability of these biomarkers across central nervous system (CNS) injuries [35–37].

The focus on SCI and TBI in this study is thus driven by their shared traumatic etiology and the established role of calpain-mediated sodium channel cleavage in their pathophysiology. SCI, with its high prevalence of spasticity (up to 75% of patients), and TBI, a leading cause of CNS trauma, both exhibit this molecular signature in

preclinical models [21,35–37], distinguishing them from non-traumatic CNS conditions such as stroke or neurodegenerative diseases, where this mechanism is not documented. By targeting these traumatic injuries, we aim to leverage this specific biomarker to address the unmet diagnostic and prognostic needs in acute CNS trauma management.

The present *SpasT-SCI-T* clinical trial protocol aims to evaluate calpain-mediated sodium channel fragments as diagnostic biomarkers for SCI and TBI and as prognostic markers for spasticity. By bridging molecular mechanisms and clinical outcomes, this study seeks to advance the management of CNS injuries and spasticity, paving the way for more accurate diagnoses, enhanced prognostic precision, and personalized therapeutic strategies.

## Methods and analysis

### Study aims

The primary objective of the SpasT-SCI-T study is to detect and quantify sodium channel fragments in blood samples from patients with SCI or TBI. These fragments will be measured at multiple post-trauma time points, ranging from 6 hours to 6 months after injury.

The secondary objectives are threefold:

1. To characterize in SCI and TBI patients the temporal kinetics of sodium channel fragments, providing insight into their dynamic expression and persistence over time.

2. To evaluate the relation between plasma fragment levels and the onset, progression, and severity of spasticity in SCI patients.

3. To compare plasma fragment levels in SCI and TBI patients with those of healthy controls, establishing baseline values and identifying injury-specific signatures.

### Study design

The SpasT-SCI-T study is a prospective, multicenter observational trial performed in France. The study employs a combination of case-control and cohort designs to address its primary and secondary objectives. The inclusion of patients will officially begin on March 17, 2025 and end on September 20, 2026. A total of 40 participants are recruited, divided into three groups:

1. **Group 1 (Healthy Controls):** Ten healthy volunteers provide baseline blood samples for comparison.

2. **Group 2 (SCI Patients):** Twenty patients with acute traumatic SCI confirmed by imaging and no associated TBI.

3. **Group 3 (TBI Patients):** Ten patients with acute TBI confirmed by imaging and no associated SCI.

The trial consists of three phases:

1. **Inclusion Phase:** Participants are enrolled into one of the three study groups based on predefined eligibility criteria.

2. **Acute Phase (0–14 days):** This phase focuses on the immediate response to trauma, capturing rapid changes in sodium channel fragment levels during the critical early period. Blood samples are collected within 6 hours post-trauma and at multiple time points (Days 1, 3, 5, 7, and 14) to monitor early biomarker kinetics.

3. **Late Follow-Up Phase (3 and 6 months):** The late follow-up phase assesses long-term changes in sodium channel fragment levels and their association with recovery trajectories and clinical outcomes, including the onset and severity of spasticity. Blood samples are collected during follow-up consultations at 3 and 6 months post-trauma. A 6-month follow-up was selected as clinical signs of spasticity typically stabilize between 2 and 6 months post-injury, with electrophysiological measures showing minimal change thereafter [38].

The CONSORT diagram in Fig 1 provides details on the timeline for enrollment, interventions, and assessments across the participant groups, while the overall study design is summarized in Fig 2.

## Ethics approval and consent

The SpasT-SCI-T study was approved by the Comité de Protection des Personnes Est III (CPP Est III) on October 7, 2024 (CPP number 2024-A00265-42). Written informed consent was obtained from all participants or their legally authorized representatives prior to inclusion. For conscious patients, consent was secured directly using the model consent form within 6 hours post-trauma. For unconscious or incapacitated patients, proxy consent was obtained from a legally authorized representative, with continuation consent sought from the patient upon recovery of decision-making capacity. In emergency cases where no proxy was available within the 6-hour inclusion window, a physician not involved in the study authorized initial inclusion, followed by written consent from the patient or representative as soon as feasible, in compliance with Article L1122-1–3 of the French Public Health Code. Consent was not waived for any participants.

## Sites and ressources

The trial spans 30 months and is performed within the Assistance Publique–Hôpitaux de Marseille (AP-HM) network, leveraging multiple specialized facilities for recruitment, data collection, and biological sample processing:

1. **Neurosurgery Department, Hôpital Nord:** Led by Professor Pierre-Hugues Roche, the Neurosurgery Department serves as the primary site for recruiting SCI and TBI patients. It specializes in acute neurological trauma management and advanced imaging (CT and MRI) to confirm eligibility. Follow-up visits at 3 and 6 months, including Modified Ashworth Scale (MAS) evaluations for spasticity in SCI patients, are coordinated here.

2. **Intensive Care Unit, Hôpital Nord:** Headed by Professor Marc Leone, the center manages sever trauma patients during the acute injury phase and ensures the inclusion of patients unable to provide immediate consent. Baseline blood samples are collected within 6 hours post-trauma, with additional samples taken during early follow-up on Days 1, 3, 5, 7 and 14.

3. **Clinical Investigation Center, Hôpital Nord:** Healthy controls are recruited and assessed at this Center. It provides a controlled environment for obtaining informed consent and baseline blood samples. Standardized protocols ensure consistency across participant groups.

4. **BioMINT Platform, INT UMR7289, Aix-Marseille University la Timone:** Biological sample processing is centralized at the BioMINT platform, part of the Institute of Neurosciences (INT UMR7289). Plasma samples are analyzed using semi-automated Western blot techniques with Pan-Nav antibodies, ensuring high-quality, standardized biomarker data. Samples are stored at -80°C until analysis and transported monthly from clinical sites.

5. **Rehabilitation Centers for Long-Term Follow-Up:** Patients transitioning to affiliated rehabilitation centers within the AP-HM network participate in long-term follow-up. These centers collaborate with the Neurosurgery Department to ensure the collection of clinical data, including MAS evaluations, at 3 and 6 months post-trauma.

## Participants

The selection of participants is based on strict inclusion and exclusion criteria (Table 1).
    Inclusion Criteria:

1. Group 1 (Healthy Controls):

• Adults (male or female) aged 18–75 years.

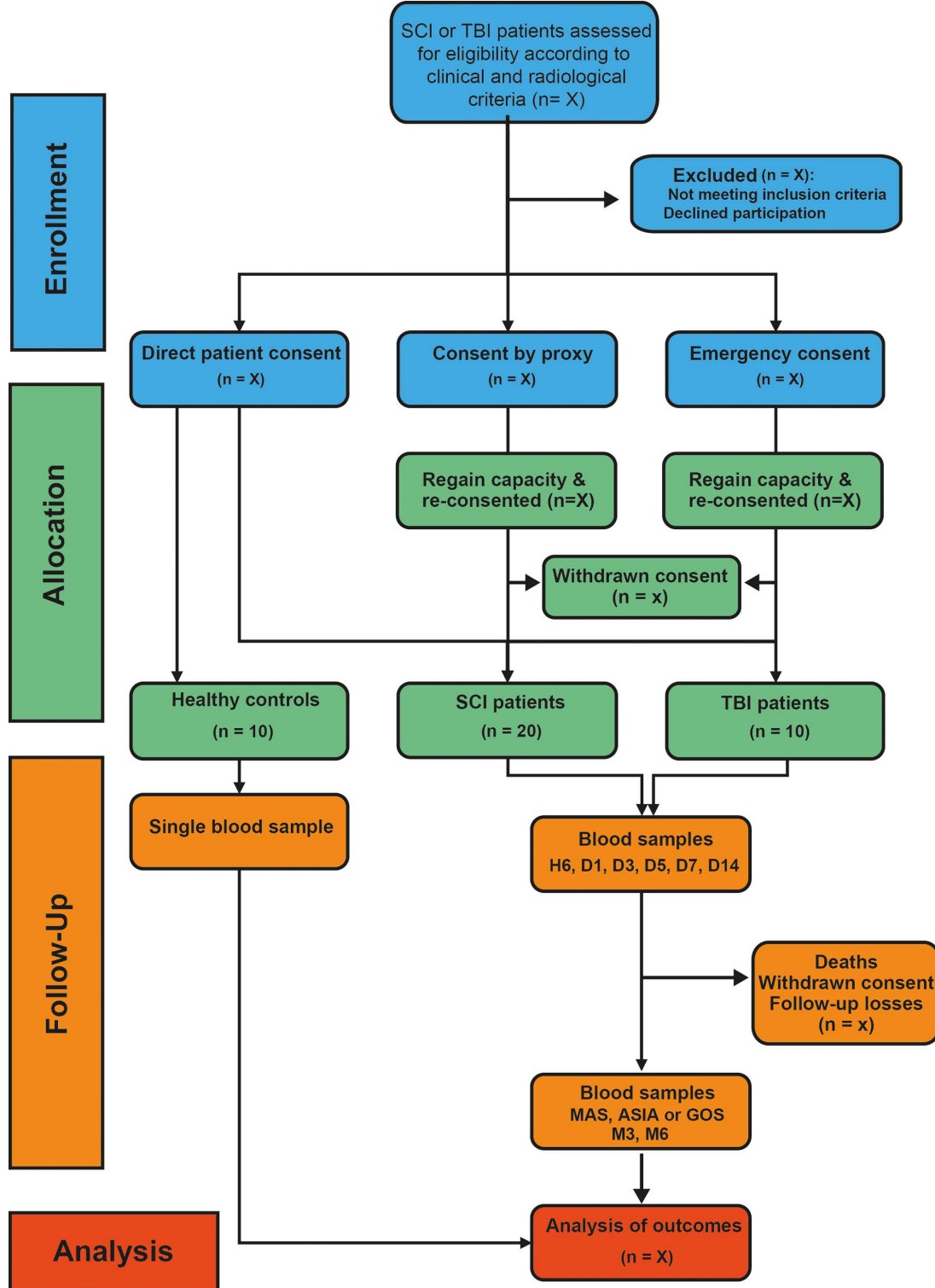

**Fig 1. CONSORT flow diagram of the SpasT-SCI-T study.** (SCI: spinal cord injury; TBI: traumatic brain injury; MAS: modified Ashworth scale; ASIA: American Spinal Injury Association; GOS: Glasgow Outcome Scale; H: hour; D: Day; M: Month). Patients under proxy or emergency consent who regain capacity are re-consented, with the option to withdraw and request retrospective data withdrawal.

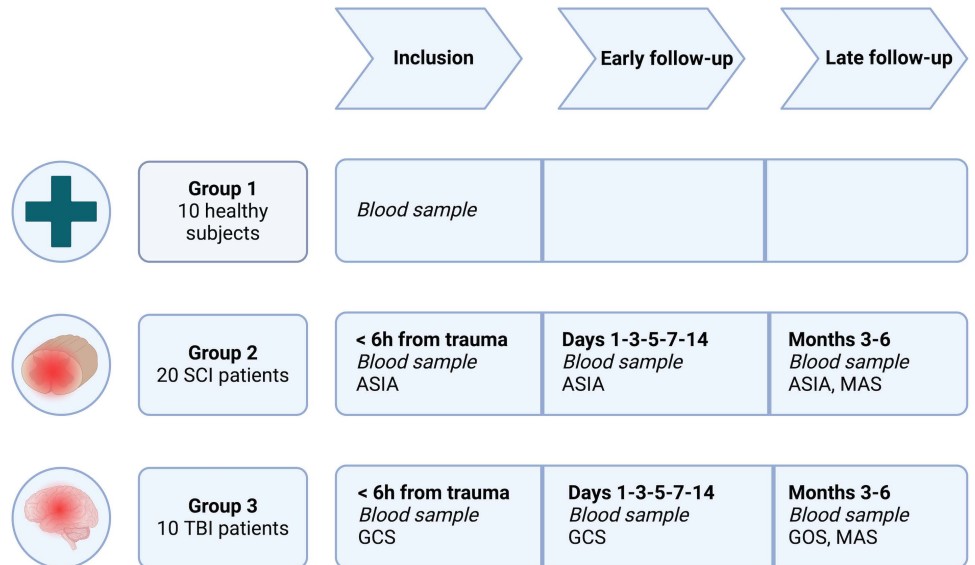

**Fig 2. Graphical overview of SpasT-SCI-T study design.** Overview of the three study groups and their respective schedules for blood sample collections and clinical assessments. Group 1 (10 healthy subjects) provides a single baseline blood sample. Group 2 (20 SCI patients) and Group 3 (10 TBI patients) undergo sampling within 6 hours post-trauma, during the early follow-up phase (days 1, 3, 5, 7, and 14), and at late follow-ups (3 and 6 months). Clinical assessments include the American Spinal Injury Association (ASIA) impairment scale, Glasgow Coma Scale (GCS), Glasgow Outcome Scale (GOS), and Modified Ashworth Scale (MAS), depending on the group. These outcome measures are defined and detailed in the "Outcomes" section.

- No history of central or peripheral nervous system trauma.

- Absence of any chronic neurological, psychiatric, or systemic diseases.

- Ability to provide informed consent prior to inclusion.

- Affiliation with a social security system.

2. Group 2 (SCI Patients):

- Adults (male or female) aged 18–75 years

- Confirmed acute SCI (C4-T12) based on imaging (CT and MRI).

- Neurological assessment using the American Spinal Injury Association Impairment Scale (ASIA): grades A to D.

- Inclusion must occur within 6 hours post-trauma.

- No concurrent TBI.

- Affiliation with a social security system.

3. Group 3 (TBI Patients):

- Adults (male or female) aged 18–75 years

- Confirmed acute TBI based on CT imaging (Marshall Grade II or higher).

- Neurological assessment using the Glasgow Coma Scale (GCS) ≤13.

**Table 1. Inclusion and exclusion criteria for study participants.**

| Criteria | Group 1: Healthy Controls | Group 2: SCI Patients | Group 3: TBI Patients |
|---|---|---|---|
| **Age (years)** | 18–75 | 18–75 | 18–75 |
| **Type of Trauma** | None | Acute spinal cord injury (C4–T12), confirmed by CT/MRI | Acute traumatic brain injury (Marshall Grade II or higher), confirmed by CT |
| **Neurological Assessment** | None | ASIA Impairment Scale (grades A to D) | Glasgow Coma Scale (GCS ≤ 13) |
| **Inclusion Timeline** | – | ≤6 hours post-trauma | ≤6 hours post-trauma |
| **Associated Conditions** | -See Exclusion Criteria | No concomitant TBI | No concomitant SCI |
| **Social Security Affiliation** | Required | Required | Required |
| **Exclusion Criteria** | - Minor, Pregnancy or breastfeeding.<br>- Pre-existing neurological disorders (e.g., multiple sclerosis, Parkinson's disease), cardiovascular or pulmonary diseases.<br>- Psychiatric illnesses or neurodegenerative conditions- Severe trauma (e.g., multi-organ failure)<br>- Inability to provide consent and absence of a legally authorized representative | -Same as Group 1 | -Same as Group 1 |

SCI, spinal cord injury; GCS, glasgow coma scale; TBI, traumatic brain injury; ASIA, American Spinal Injury Association.

- Inclusion must occur within 6 hours post-trauma.

- No concurrent SCI.

- Affiliation with a social security system.

  Exclusion Criteria (All Groups):

- Pregnancy or breastfeeding.

- Pre-existing neurological disorders, such as multiple sclerosis or Parkinson's disease.

- History of psychiatric illness or neurodegenerative conditions.

- Concurrent severe trauma affecting study participation (e.g., multi-organ failure).

- Inability to provide consent, and absence of a legally authorized representative.

- Pre-existing cardiovascular or pulmonary diseases.

Informed consent is obtained from all participants using the model consent form and related documents after verifying their eligibility according to the inclusion and exclusion criteria. Recognizing the urgency of inclusion and the acute medical care context, the consent process is tailored to accommodate the specific needs of SCI and TBI patients. The following procedures ensure compliance with ethical and legal requirements while prioritizing participant autonomy and rights:

1. **For conscious patients** capable of decision-making, informed consent is obtained directly within the 6-hour inclusion window. The investigator or clinical team provides the patient with a detailed information sheet outlining the study's objectives, procedures, potential risks and benefits. Patients are given the opportunity to ask questions and are encouraged to make an informed decision. Once all concerns are addressed, they sign the consent form, confirming their voluntary participation.

2. **For unconscious or incapacitated patients**, a legally authorized representative (e.g., a close relative) is approached to provide proxy consent. The representative is provided with the same detailed information about the study, and their consent is carefully documented. Once the patient regains the capacity to decide, they are fully informed about their participation, following the same process as for conscious patients. This involves a prompt visit from the investigator or a trained clinical team member, who provides a detailed information sheet and verbally explains the study's objectives, procedures, risks, benefits, and their prior inclusion under emergency or proxy consent. Patients are given ample time to ask questions and consider their options. Written continuation consent is then requested to confirm their voluntary agreement to remain in the study. If they decline continuation consent, they may withdraw from the study and can request retrospective withdrawal of data collected prior to regaining capacity, in accordance with Article L1122-1–1 of the French Public Health Code and GDPR. Such data will be destroyed unless already anonymized and aggregated for analysis. These procedures comply with Article L1122-1–3 of the French Public Health Code, ensuring adherence to legal and ethical standards for proxy and emergency consent.

3. **Emergency Consent:** In cases where the patient is unconscious and no proxy is available within the inclusion time-frame (6 hours post-trauma), a physician not involved in the study may authorize emergency consent. This process allows for the patient's inclusion while ensuring that: 1) Consent is obtained as soon as possible from the patient or their representative once circumstances allow. Once the patient regains capacity or a relative is available, they are fully informed of their inclusion and written consent is requested, in compliance with Article L1122-1–3 of the French Public Health Code. The re-consent and withdrawal process outlined above applies in these cases, ensuring patients retain full autonomy over their participation and data.

4. For **healthy volunteers**, written informed consent is obtained prior to any study-related procedures. The consent process ensures they understand the study's objectives, any potential risks, and their right to withdraw at any time without consequences.

All participants, regardless of group, continue to receive standard medical care, including routine imaging, rehabilitation, and clinical follow-ups, without interference from study procedures. Withdrawal from the study, whether initiated by the participant or deemed necessary for medical reasons, does not impact their access to or quality of standard medical care.

### Intervention procedures

The SpasT-SCI-T trial outlines specific intervention procedures for each participant group. These interventions encompass blood sampling, imaging, and clinical assessments aimed at investigating sodium channel fragments as potential biomarkers and correlating their dynamics with clinical outcomes. All procedures are rigorously standardized to ensure reproducibility and protocol adherence.

**Group 1: Healthy subjects.** Healthy participants are selected from a pre-screened volunteer registry. Upon enrollment, they are invited to the Clinical Investigation Center at Hôpital Nord at their convenience. During the inclusion visit a single venous blood sample of 4 mL is collected in BP100 tubes containing protease inhibitors to prevent biomarker degradation. The sample is processed within 2 hours of collection by centrifugation at 1500g to extract plasma, which is then stored at -80°C until analysis at the BioMINT platform. No follow-up visits or additional interventions are required for this group, ensuring minimal participant burden.

**Group 2: Spinal Cord Injury (SCI) patients.** SCI patients are prospectively recruited from the Intensive Care Unit or Neurosurgical Department at Hôpital Nord within 6 hours post-trauma. Upon admission, interventions for these patients begin with clinical and imaging assessments. A computed tomography (CT) scan is performed on admission to confirm vertebral fractures or spinal column instability. Magnetic resonance imaging (MRI) provides detailed characterization of the SCI, documenting patterns such as contusion, compression, hemorrhage, or edema. Injury severity is further classified using the AOSpine fracture classification [39] and BASIC scoring system [40,41].

Baseline neurological status is documented using the American Spinal Injury Association (ASIA) Impairment Scale, which evaluates the severity and level of the SCI [39,42]. Additional documentation includes respiratory and hemodynamic stability, renal clearance, liver function, and hemoglobin levels.

Blood sampling begins with a 4 mL venous sample collected within the 6 hours of trauma. Additional samples are collected during the acute phase on days 1, 3, 5, 7 and 14 to monitor early biomarker kinetics. Further samples are obtained at 3 months (M3) and 6 months (M6) post-trauma, offering insights into long-term biomarker dynamics. As part of the follow-up interventions, at M3 and M6, spasticity is assessed using the **Modified Ashworth Scale** (MAS, Table 2), alongside detailed evaluations of neurological recovery and residual deficits. These assessments are complemented by documentation of neurological recovery and residual deficits, enabling correlations between clinical outcomes and sodium channel fragment levels.

**Group 3: Traumatic Brain Injury (TBI) patients.** Interventions for TBI patients are tailored to intracranial trauma and are similar in structure to those for SCI patients, with modifications to address brain injury specifics. A whole-body CT scan is performed upon admission to confirm intracranial hemorrhage and exclude SCI, ensuring group specificity. Neurological status is assessed using the Glasgow Coma Scale (GCS) to document the severity of brain injury [43]. Trauma-related data, including respiratory and hemodynamic stability, renal clearance, liver function, and hemoglobin levels, are recorded. Blood sampling begins with a 4 mL venous sample collected within the 6 hours of trauma during the routine biological workup. Additional samples are obtained during the acute phase at Days 1, 3, 5, 7 and 14 to capture early biomarker dynamics. Further samples are collected during late follow-up visits at M3 and M6 to evaluate the longer-term biomarker behavior. General neurological assessments are performed at M3 and M6 to evaluate recovery. These outcomes are correlated with sodium channel fragment levels to explore potential biomarker-based prognostication.

For SCI and TBI patients (Groups 2 and 3), detailed records of all medical treatments administered during hospitalization will be maintained. This includes, but is not limited to: anticoagulation therapy, antibiotic regimens, anti-spastic medications, and pain management protocols. This documentation allows for the evaluation of potential confounding factors and ensures that biomarker changes can be interpreted in the context of treatment.

## Outcomes

**The primary outcome** of the study is the detection and quantification of sodium channel fragments in blood samples. These fragments are quantified using a semi-automated Western blot technique employing the commercial Pan-Nav antibody, which is validated for its high sensitivity in detecting α-subunit cleavage products generated by calpain-mediated proteolysis. Sodium channel fragment levels are compared across SCI, TBI and healthy participants to identify injury-specific biomarker signatures. These data aim to provide valuable insights into the molecular mechanisms underlying SCI and TBI.

**Table 2. Modified Ashworth scale for spasticity assessment [44].**

| Modified Ashworth score | Clinical findings |
|---|---|
| 0 | No increase in muscle tone |
| 1 | Slight increase in muscle tone, with a catch and release or minimal resistance at the end of the range of motion when an affected part(s) is moved in flexion or extension |
| 1+ | Slight increase in muscle tone, manifested as a catch, followed by minimal resistance through the remainder (less than half) of the range of motion |
| 2 | A marked increase in muscle tone throughout most of the range of motion, but the affected part(s) are still easily moved |
| 3 | Considerable increase in muscle tone, passive movement difficult |
| 4 | Affected part(s) rigid in flexion or extension |

The **secondary outcomes** include:

1. **Temporal Dynamics of Sodium Channel Fragments:** The temporal kinetics of sodium channel fragments are analyzed to characterize their dynamic expression and persistence during the acute and chronic phases of injury. Longitudinal changes in fragment levels are assessed from the baseline (acute phase) through follow-up visits at 3 months (M3) and 6 months (M6). These data are related with key clinical and functional outcomes to evaluate the role of sodium channel fragments as potential prognostic biomarkers. For SCI patients, changes in neurological function are monitored using the American Spinal Injury Association (ASIA) Impairment Scale, which provides detailed assessments of sensory and motor function recovery. For TBI patients, functional recovery is evaluated using the Glasgow Outcome Scale (GOS) and the more granular Glasgow Outcome Scale-Extended (GOSE), which capture subtle variations in recovery trajectories. For both groups, quality of life is assessed using the Short Form Health Survey (SF-36), a comprehensive tool that evaluates health-related quality of life across physical, mental, and social dimensions. These correlations aim to establish the predictive value of sodium channel fragment levels for neurological recovery, functional outcomes, and overall quality of life, providing a deeper understanding of the molecular mechanisms underlying injury progression and recovery.

2. **Spasticity and Biomarker Levels in SCI Patients:** The relationship between sodium channel fragment levels and spasticity is specifically investigated in SCI patients. Spasticity is assessed during follow-up visits at M3 and M6 using the Modified Ashworth Scale (MAS), a widely used tool for grading muscle tone. The relationship between MAS scores and fragment levels is evaluated using correlation analyses. This correlation is expected to provide evidence of the prognostic utility of sodium channel fragments in predicting the onset and severity of spasticity. These analyses aim to establish the prognostic utility of sodium channel fragments in predicting the onset, progression, and severity of spasticity, providing a molecular link between injury mechanisms and clinical outcomes. Such insights may inform personalized therapeutic strategies for managing spasticity in SCI patients.

3. **Injury Severity and Biomarker Associations:** The study investigates the association between sodium channel fragment levels and injury severity in both SCI and TBI patients. For SCI patients, neurological severity is assessed using the ASIA Impairment Scale, which evaluates the level and completeness of spinal cord injury through sensory and motor function tests. For TBI patients, the severity of brain injury is determined using the GCS, which provides a standardized assessment of consciousness based on eye, verbal, and motor responses. By correlating sodium channel fragment levels with these established severity metrics, the study aims to elucidate the relationship between molecular biomarkers and clinical injury severity, enhancing the understanding of the pathophysiology of CNS trauma and informing future diagnostic approaches.

### Biomarker levels

Sodium channel fragment levels are quantified using the Jess system (ProteinSimple, Bio-techne), a capillary-based automated Western blot platform. Plasma samples are resuspended in 0.1X Sample buffer (SimpleWestern, Bio-techne). Protein concentrations are determined using the DC Protein Assay (Biorad) to ensure consistent loading across all samples. Samples are prepared without reducing agents or heat denaturation to preserve protein integrity. Prepared plasma samples (3 µL) are loaded into the Jess assay plate along with controls, standards, and molecular weight ladders. Proteins are loaded into the capillary automatically (12–230 kDa range), electrophoresed and separated by size. A monoclonal anti-pan Nav antibody (Sigma-Aldrich, CAT#S8809, 1:500) targeting the α-subunit of human sodium channels is used for detection. This antibody, raised against the conserved intracellular III-IV loop peptide 'CTEEQKKYYNAMKKLGSKK,' was validated for specificity by immunoblotting in HEK293 cells transfected with Nav channels, showing selective detection of the native channel (~250 kDa) and its cleavage fragments, with no signal in untransfected controls [21]. The conserved sequence

minimizes cross-reactivity. Chemiluminescence is used for signal detection. Quantitative data are captured by automated software, and protein band intensities are normalized to either a housekeeping protein or total protein for variability control. The native sodium channel (~250 kDa) and its lower molecular weight fragments are identified and quantified based on band intensity, facilitating the comparison of fragment expression across groups.

## Spasticity evaluations

Spasticity is assessed at M3 and M6 follow-up visits using the Modified Ashworth Scale (MAS), which grades muscle tone on a scale from 0 (no increase in muscle tone) to 4 (rigid muscle tone) [44]. This score measures the resistance perceived by the rater when passively moving a joint through its full range of movement. All evaluations are performed by a single, trained physiotherapist to minimize inter-observer variability. Associated phenomena, such as clonus and spasms, are also recorded to provide a comprehensive spasticity profile. Factors influencing variability (e.g., patient positioning, medication, and pain) are meticulously documented.

## Neurological Evaluations (GCS, ASIA)

**The Glasgow Coma Scale (GCS)** is a widely used, standardized tool for assessing consciousness in patients with TBI. It evaluates three key components: eye-opening, verbal, and motor responses, providing a rapid and reliable measure of neurological status. The Eye-Opening Response is scored from 4 to 1: 4 for spontaneous eye opening, 3 for a response to verbal stimuli, 2 for a response to painful stimuli, and 1 if there is no response. The Verbal Response evaluates the orientation and ability to communicate, scoring 5 for coherent and oriented speech, 4 for confusion, 3 for inappropriate words, 2 for incomprehensible sounds, and 1 for no vocalization. The Motor Response measures movement in response to commands or stimuli, scoring 6 for obeying commands, 5 for localizing pain, 4 for withdrawing from pain, 3 for abnormal flexion (decorticate posturing), 2 for abnormal extension (decerebrate posturing), and 1 for no response. The total GCS score ranges from 3 (deep coma) to 15 (full consciousness), allowing clinicians to classify the severity of brain injury and monitor changes in neurological status over time.

**The American Spinal Injury Association (ASIA) Impairment Scale** is a standardized framework for evaluating and classifying the severity of SCI [39,42]. It assesses sensory and motor functions across key dermatomes and myotomes, as well as sacral sparing, to determine the neurological level and completeness of the injury. Sensory evaluation tests light touch and pinprick sensations across 28 dermatomes bilaterally, from C2 to S4/S5. A cotton swab is used for light touch, and a safety pin for pinprick testing, with scores assigned as 2 for normal sensation, 1 for impaired sensation, and 0 for absent sensation. Special attention is given to the S4/S5 dermatome, as sacral sparing is critical for determining whether the injury is complete or incomplete. Motor function is evaluated by testing strength in 10 muscle groups corresponding to specific myotomes, ranging from elbow flexion (C5) to ankle plantar flexion (S1), scored from 5 (normal strength) to 0 (no contraction). Sacral motor function is further assessed by voluntary contraction of the external anal sphincter. The ASIA Impairment Scale assigns grades based on the completeness of the injury: Grade A indicates complete injury with no sensory or motor function preserved in sacral segments, Grades B-D reflect varying degrees of incomplete injuries with some preserved sensory or motor function below the injury level, and Grade E signifies normal sensory and motor function. The neurological level of injury is defined as the lowest spinal segment with intact sensory and motor function bilaterally.

## Functional Scales (GOS, GOSE, SF-36)

The **Glasgow Outcome Scale (GOS)** is a widely recognized tool for evaluating functional recovery following TBI [45]. It categorizes outcomes into five levels: death, vegetative state, severe disability, moderate disability, and good recovery. The GOS provides a high-level summary of a patient's ability to reintegrate into daily life post-injury. To enhance its sensitivity, the **Glasgow Outcome Scale-Extended (GOSE)** subdivides each GOS category into upper and lower levels,

offering a more nuanced evaluation of recovery trajectories. This refined scale is particularly beneficial for detecting subtle differences in functional outcomes, making it a valuable tool in clinical trials. Both the GOS and GOSE are exclusively applied to TBI patients in this study to monitor recovery.

The **Short Form Health Survey (SF-36)** is a generic quality-of-life questionnaire used for both SCI and TBI patients. It evaluates health-related quality of life (HRQoL) across eight domains, including physical functioning, pain, general health, vitality, social functioning, and mental health [46]. The SF-36 captures the broader impact of injuries and treatments on physical, emotional, and social well-being. Administered during follow-up visits, it complements the GOS and GOSE for TBI patients by offering additional insights into their quality of life. For SCI patients, it serves as a critical tool for assessing long-term functional impairments and evaluating the effectiveness of rehabilitation strategies.

### Pain assessments (VAS, DN4)

Pain assessments are critical for evaluating quality of life and rehabilitation in SCI and TBI patients. The **Visual Analogue Scale (VAS)** is a simple yet effective tool for quantifying pain intensity. Patients rate their pain on a 10-centimeter line, with anchors at "no pain" (0 cm) and "worst imaginable pain" (10 cm). This approach enables longitudinal tracking of pain severity, offering valuable insights into the effectiveness of therapeutic interventions over time. The **DN4 questionnaire (Douleur Neuropathique 4 Questions)** is specifically designed to identify neuropathic pain, which is prevalent among SCI patients [47]. It comprises 10 items: 7 symptom-related questions (e.g., burning, tingling, electric shocks) and 3 clinical examination items (e.g., hypoesthesia, allodynia). A score of ≥4 indicates the presence of neuropathic pain. Together, the VAS and DN4 provide a comprehensive evaluation of pain intensity and type, enabling targeted therapeutic strategies and effective monitoring of patient progress.

### Sphincter functions

Sphincter function is a crucial outcome in SCI patients, as SCI often compromise bladder and bowel control, significantly affecting quality of life. Assessments include bladder management (permanent or intermittent catheterization, normal urination) and bowel function (fecal incontinence, constipation), focusing on their impact on quality of life.

The **Table 3** summarizes the main data collected from the patients in the case report forms, from their inclusion in the study to the 6-month follow-up consultation.

### Justification of sample size

The study includes a total of 40 participants distributed as follows: 20 patients with spinal cord injury (SCI), 10 patients with traumatic brain injury (TBI), and 10 healthy controls. The selection of 20 SCI patients is based on the annual admission rates for SCI at the study institution, ensuring a representative sample of the population served. The inclusion of 10 TBI patients is justified to account for potential confounding factors arising from the overlapping but distinct injury mechanisms of SCI and TBI. Furthermore, 10 healthy controls are included to provide a robust baseline for sodium channel fragment levels, facilitating a balanced comparison across the three groups.

This sample size is sufficient to detect meaningful trends and associations between sodium channel fragments and clinical outcomes, including spasticity, while maintaining feasibility within the scope of a pilot study. The findings will generate preliminary evidence regarding the diagnostic and prognostic utility of sodium channel fragments in neurological injuries and will guide the design of larger-scale clinical trials.

Given the exploratory nature of this study, no formal interim analyses are planned. As the trial involves minimal-risk procedures, such as blood sampling, no pre-specified stopping rules have been established. In the event of significant deviations from the protocol or unforeseen safety concerns, the trial steering committee will consult the Ethics Committee to determine whether adjustments or study termination are necessary. All such decisions will be transparently documented and communicated to relevant stakeholders, including regulatory authorities, to ensure compliance with ethical standards.

## Randomisation, blinding

The SpasT-SCI-T trial is an observational study employing a case-control and cohort design; therefore, randomization is not applicable. Participants are allocated to groups based on predefined eligibility criteria rather than random assignment. Consequently, sequence generation, allocation concealment, and implementation of randomization do not apply to this trial. To mitigate bias and maintain data integrity, laboratory personnel conducting biochemical analyses remain blinded to group allocation. This blinding ensures that sodium channel fragment quantification is performed without preconceived assumptions about the samples, enhancing the reliability and validity of the results.

## Statistical methods

The primary objective of this study is to quantify sodium channel fragments in blood samples and compare their levels across the three participant groups: healthy controls (Group 1), SCI patients (Group 2) and TBI patients (Group 3). To address this objective, statistical analyses will begin with group comparisons.

1. **Group Comparisons:** For normally distributed data, one-way analysis of variance (ANOVA) will be employed to identify differences in sodium channel fragment levels among the three groups. If data are not normally distributed, the Kruskal-Wallis test will be applied as a non-parametric alternative. Post-hoc analyses will follow these tests to assess pairwise comparisons, using Tukey's test for ANOVA or the Mann-Whitney U test for non-parametric data.

2. **Effect Size Calculation:** Effect sizes (e.g. Cohen's d for t-tests or $\eta^2$ for ANOVA) will be calculated to quantify the magnitude of differences between groups.

For secondary outcomes, analyses will focus on the temporal kinetics of sodium channel fragments and their correlation with clinical outcomes.

1. **Temporal Kinetics:** Changes in fragment levels over time, measured at baseline (6 hours), during the acute phase (Days 1, 3, 5, 7), and at follow-up (Months 3 and 6), will be analyzed using repeated-measures ANOVA for normally distributed data. For non-normally distributed data, the Friedman test will be used. Bonferroni corrections will be applied during multiple comparisons.

2. **Spasticity Correlation:** The relationship between sodium channel fragment levels and spasticity severity in SCI patients will be assessed using Spearman's rank correlation coefficients, with spasticity evaluated at Months 3 and 6 using the Modified Ashworth Scale (MAS).

3. **Subgroup Analysis (SCI Severity):** Subgroup analyses will investigate whether sodium channel fragment levels differ by injury severity. In SCI patients, severity will be categorized based on the American Spinal Injury Association (ASIA) Impairment Scale grades (A–D). Similarly, spasticity onset will be analyzed as a binary outcome (presence vs. absence). For TBI patients, the Glasgow Coma Scale (GCS) will be used to classify injury severity, enabling comparisons of sodium channel fragment levels across defined severity subgroups.

Additional statistical methods will include multivariable linear regression models to adjust for potential confounders, including age, sex, injury severity (ASIA grades A–D for SCI, GCS scores for TBI), and concurrent medications (antispasmodics, analgesics, antiepileptics) as primary factors, as well as additional variables from Table 3 such as comorbidities (diabetes, organ failure), trauma context (type of accident), and biological parameters (hemoglobin, renal clearance) where statistically significant or clinically relevant, informed by prior literature [14,21]. This ensures that observed associations between sodium channel fragment levels and outcomes such as spasticity are robust and not influenced by baseline characteristics. Receiver operating characteristic (ROC) curves will be generated to identify threshold levels of sodium channel fragments that differentiate SCI from TBI or predict spasticity onset. To address missing data, multiple imputation methods will be applied, minimizing bias and preserving the validity of the analyses.

**Table 3. Main data collected from the patients in the case report forms, from their inclusion in the study to the 6-month follow-up consultation (ASIA: American Spinal Injury Association; GCS: Glasgow Coma Scale; BASIC: Brain and Spinal Injury Center [41]; AOSpine [39]; AIS: Abbreviated Injury Score; ISS: Injury Severity Score [48]; SAPS II: Simplified Acute Physiology Score II [49]; SOFA: Sepsis related Organ Faillure Assessment [50]; Kdigo [51]; modified Ashworth scale [44]; DN4 [47]; GOS: Glasgow Outcome Scale; GOSE: Glasgow Outcome Scale Extended [45]; SF36: Short Form 36 Health Survey [46]).**

| Inclusion | Hospital discharge | 3 and 6-month follow-up |
|---|---|---|
| Background:<br>-Gender<br>-Age<br>-Height<br>-Weight<br>Medical history:<br>-Organ failure: heart, lung, kidney<br>-Diabetes and chronic diseases<br>-Cancer<br>-Addiction: smoking, alcoholism | Spinal surgery:<br>-Spinal stabilization<br>-Spinal cord decompression<br>-Levels<br>Cranial surgery:<br>-Intracranial lesion evacuation (craniotomy or craniectomy)<br>-External ventricular drainage<br>-Intracranial pressure monitoring<br>Other surgery | Motricity:<br>-Bed rest<br>-Wheelchair use<br>-Walking with assistance<br>-Slow walking without assistance<br>-Walking normally |
| Usual treatment:<br>-Antiepileptic<br>-Anticoagulant<br>-Antihypertensive<br>-Antidiabetic… | Additional treatments:<br>-Analgesics<br>-Antispasmodics<br>-Antiepileptics<br>-Antibiotics... | Additional treatments:<br>-Analgesics<br>-Antispasmodics<br>-Antiepileptics<br>-Antibiotics... |
| Trauma:<br>-Date and time<br>-Inclusion time<br>-Type of accident | Intensive care stay:<br>-Duration: stay, mechanical ventilation, sedation, intracranial pressure monitoring<br>-Complications: refractory intracranial hypertension, infections (with treatment>72 hours), septic shock, acute respiratory distress syndrome, acute renal failure (Kdigo>3), cardiorespiratory arrest | Rehabilitation stay:<br>-Medical complication<br>-Specific support |
| Neurological evaluation:<br>-ASIA score<br>-GCS score | Neurological evaluation:<br>-ASIA score<br>-GCS score | Neurological evaluation:<br>-ASIA score<br>-GCS score |
| Clinical evaluation:<br>-Heart rate<br>-Blood pressure<br>-Respiratory rate<br>-Oxygen saturation | Spasticity:<br>-Location<br>-Modified Ashworth scale<br>-Spasm, clonus<br>-Treatment: antispastic, antiepileptic | Spasticity:<br>-Location<br>-Modified Ashworth scale<br>-Spasm, clonus<br>-Abnormal posture<br>-Treatment: antispastic, antiepileptic |
| Pre-hospitalization management:<br>-Cardiopulmonary arrest<br>-Hypovolemic shock<br>-Fluid restoration<br>-Vasopressor treatment<br>Spinal cord injury:<br>-Level<br>-BASIC score<br>-AOSpine vertebral fracture type<br>Traumatic brain injury:<br>-Intracranial lesions<br>-Rotterdam score<br>Trauma scores:<br>-AIS and ISS<br>-IGS2 and SOFA | Pain:<br>-Location, duration, frequency<br>-Visual analogue scale<br>-DN4<br>-Treatment: analgesic, antiepileptic<br>Sphincter functions:<br>-Permanent bladder catheterization<br>-Iterative bladder catheterization<br>-Normal urination<br>-Fecal incontinence<br>-Constipation | Pain:<br>-Location, duration, frequency<br>-Visual analogue scale<br>-DN4<br>-Treatment: analgesic, antiepileptic<br>Sphincter functions:<br>-Permanent bladder catheterization<br>-Iterative bladder catheterization<br>-Normal urination<br>-Fecal incontinence<br>-Constipation<br>Functional scales:<br>-GOS, GOSE<br>-SF36 |
| Biology:<br>-Renal clearance<br>-Hepatic function<br>-Hemoglobin<br>-Lactate | Biology:<br>-Renal clearance<br>-Hepatic function<br>-Hemoglobin | Biology:<br>-Renal clearance<br>-Hepatic function<br>-Hemoglobin |

Continuous variables will be expressed as means with standard deviations for normally distributed data or medians with interquartile ranges for non-normally distributed data. Categorical variables will be summarized as frequencies and percentages. For all primary and secondary analyses, a two-tailed p-value < 0.05 will be considered statistically significant. Patients who die before the 3-month follow-up will be excluded from spasticity-related analyses due to insufficient time for spasticity development, but all data collected up to the time of death will be included in secondary analyses, such as demographic characterization and early biomarker levels.

## Ethical approval and dissemination policy

**Research ethics approval.** The SpasT-SCI-T clinical trial adheres to the Declaration of Helsinki and the French Public Health Code. It received approval from the Comité de Protection des Personnes Est III (CPP number 2024-A00265-42) on October 7, 2024, and authorization from the National Agency for Medicines and Health Products (ANSM) on July 10, 2017. The trial is registered on Clinicaltrials.gov (NCT06532760). As a category 2 interventional study under French law, it involves minimal risk. Reporting follows the SPIRIT guidelines, with SPIRIT and CONSORT checklists provided (S1 and S2 Checklists).

**Protocol amendments.** Any modifications to the trial protocol, including changes to eligibility criteria, outcome measures, or analytical methods, will be promptly communicated to all relevant stakeholders. These include the investigators, the Ethics Committee (CPP), regulatory authorities such as the ANSM (Agence Nationale de Sécurité du Médicament), and trial participants, when applicable. Updates will also be reflected in trial registries (e.g., ClinicalTrials. gov) and disseminated through publications and conference presentations as needed. All amendments will undergo prior review and approval by the CPP to ensure continued ethical compliance and transparency.

**Data confidentiality and management.** Participant confidentiality is prioritized throughout the trial. All data are pseudonymized, and unique patient identifiers are used for sample tracking and clinical data management. Personal identifying information will not be associated with biological samples or research databases. Data will be securely stored in a password-protected electronic Case Report Form (eCRF) developed using REDCap, hosted on encrypted servers at Assistance Publique–Hôpitaux de Marseille (AP-HM). Access to the eCRF is restricted to authorized personnel, including the principal investigators and the designated data management team, through encrypted login credentials. Data integrity is ensured through daily backups, and all procedures comply with the European General Data Protection Regulation (GDPR). At the conclusion of the trial, all biological samples will be destroyed, and the pseudonymized data will be archived securely for 15 years, as mandated by French regulations.

**Ancillary and post-trial care.** As the trial involves minimal-risk procedures, participants are not expected to require ancillary or post-trial care related to their involvement. In the event of harm directly resulting from study participation, liability insurance provided by the sponsor ensures coverage. Participants will retain access to their treating physicians for medical concerns during and after the trial.

**Dissemination plan.** The results of the SpasT-SCI-T clinical trial will be disseminated through peer-reviewed journals, conference presentations, and publicly accessible databases. The findings will adhere to CONSORT guidelines to ensure transparency and rigor in reporting. Participants who express interest in the study's results will be provided with an aggregated summary by the investigators, in compliance with French regulatory requirements. To maintain confidentiality, no individual results will be disclosed. Authorship of publications will follow the guidelines of the International Committee of Medical Journal Editors (ICMJE), ensuring that all contributors to the study design, conduct, analysis, and reporting are appropriately credited.

**Baseline data.** Baseline data were systematically collected to characterize the study population and ensure comparability across the three participant groups (healthy controls, SCI patients, and TBI patients). Key baseline variables are presented in a detailed summary table to promote transparency and support balanced comparisons between groups.

## Discussion

### Rationale

The present clinical trial aims to validate sodium channel fragments generated by calpain as biomarkers for assessing injury severity and prognosis in SCI and TBI, and to predict spasticity outcomes. CNS injuries initiate a cascade of molecular events, including calcium overload and calpain activation, leading to the proteolysis of Nav1.6 sodium channels (Fig 3). These cleavage products emerge as detectable blood biomarkers, offering a window into injury severity and secondary complications like spasticity. The SpasT-SCI-T trial builds on evidence linking calpain activity to spasticity severity [21], hypothesizing that sodium channel fragments can serve as reliable prognostic markers for functional outcomes in SCI and TBI. This aligns with the growing need for biomarkers to enhance CNS injury management, from understanding disease progression to guiding clinical decisions [14,15,52].

### Comparison with previous biomarkers

Previous studies have explored calpain-mediated α-II spectrin fragments as indicators of SCI severity [52]. However, their utility is limited by several factors: spectrin is also cleaved by caspase-3 during apoptosis, reducing its specificity to

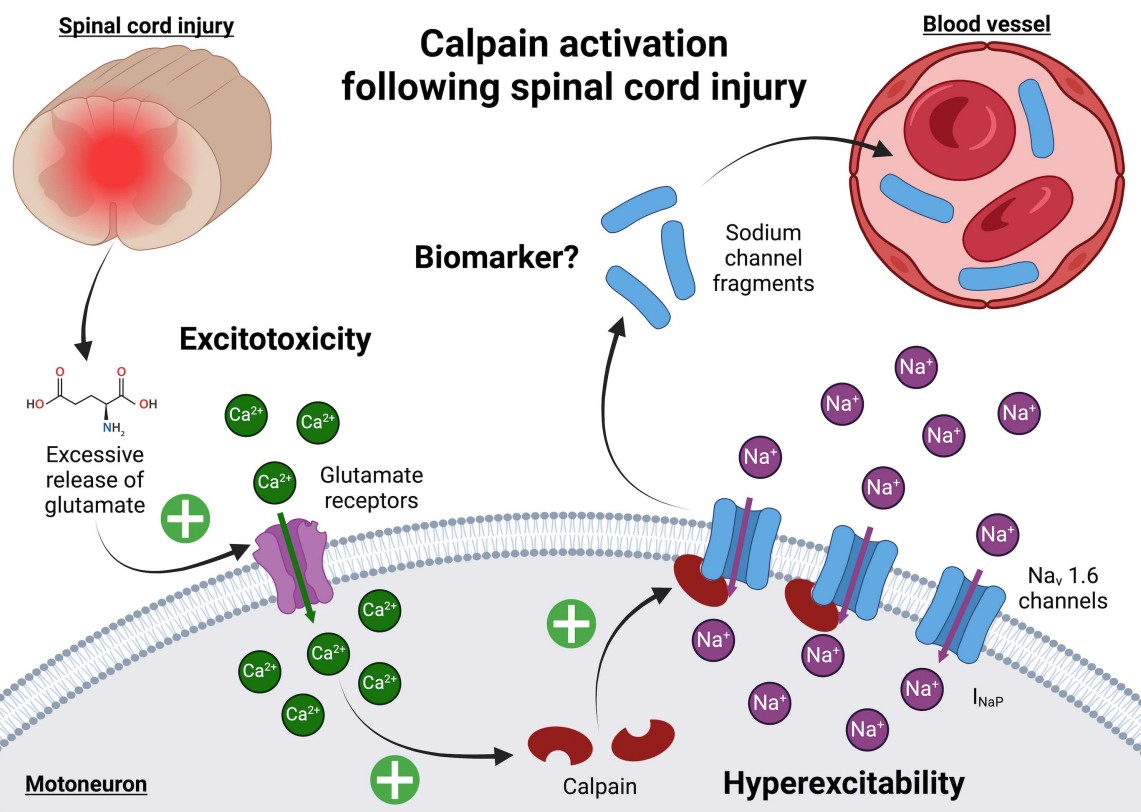

**Fig 3. Calpain activation following spinal cord injury.** Traumatic injury to the spinal cord triggers glutamate release into the extracellular space, causing a sustained increase in intracellular calcium concentrations in motoneurons. This calcium influx activates calpain, a calcium-dependent protease, which cleaves Nav1.6 voltage-gated sodium channels. The cleavage results in an upregulation of persistent sodium currents ($I_{NaP}$), contributing to motoneuron hyperexcitability and the development of spasticity. Additionally, fragments of cleaved sodium channels accumulate in the sublesional spinal cord and can be detected in the bloodstream. These fragments represent a promising biomarker for assessing SCI severity and predicting spasticity outcomes.

calpain activity [53]; its expression is transient, restricting its use to the acute phase [15]; and its abundance in red blood cells complicates interpretation in polytrauma cases [52]. In contrast, sodium channel fragments offer distinct advantages. Exclusively expressed in the nervous system and resistant to caspase-3 cleavage [37], they provide greater specificity. Moreover, preliminary data suggest they persist beyond the acute phase, enabling long-term monitoring of spasticity and recovery, attributes that address critical gaps in current biomarker options.

## Clinical implications

The potential of sodium channel fragments as biomarkers could transform SCI and TBI management. Unlike clinical assessments and imaging, which often lack sensitivity for molecular changes or long-term predictions [4–6], these fragments could offer real-time insights into injury dynamics and therapeutic efficacy. This is crucial given that spasticity affects up to 75% of SCI patients, significantly impacting their quality of life and rehabilitation outcomes [7,8]. Early identification of at-risk patients could enable targeted interventions in the acute phase, potentially mitigating spasticity severity and improving outcomes. The inclusion of TBI patients in this study further highlights a shared mechanism across CNS injuries, suggesting sodium channel fragments as versatile biomarkers for neural trauma.

## Strengths and limitations

This study uniquely explores sodium channel fragments, providing an innovative and specific biomarker linked directly to calpain-mediated pathophysiology common to SCI and TBI. Unlike previous biomarkers, sodium channel fragments offer neuronal specificity, sustained temporal expression suitable for both acute and chronic phases, and potential for wide applicability in traumatic CNS conditions. Despite their promise, challenges remain before clinical adoption. Patient factors like demographics and comorbidities may affect fragment levels, requiring larger studies to define normative ranges and thresholds. Standardized sample processing is also essential for reproducibility. Integrating biomarker analysis into clinical workflows demands infrastructure and validation efforts.

## Future directions

Future research should focus on combining biomarkers with advanced imaging techniques to enhance diagnostic precision and prognostic modeling. A multimodal approach incorporating molecular biomarkers and imaging modalities could refine injury assessment and personalize treatment strategies, advancing CNS trauma care.

## Conclusions

The SpasT-SCI-T trial represents a critical advance in linking molecular mechanisms to clinical outcomes in SCI and TBI. By establishing sodium channel fragments as robust biomarkers, this study has the potential to modernize CNS trauma management, facilitating personalized therapeutic approaches and improving long-term recovery prospects. Despite remaining challenges, its findings promise to usher in a new era of biomarker-driven neurology, propelling precision medicine forward in CNS disorder care.

## Others information

**Registration**: The SpasT-SCI-T clinical trial has been registered on the ClinicalTrials.gov database under the identifier NCT06532760.

 **Protocol Availability:** The full trial protocol for the SpasT-SCI-T study is publicly available on ClinicalTrials.gov under the identifier NCT06532760. The full trial protocol, including all details regarding the study design, objectives, methodology, and ethical considerations are described in the present paper.

## Supporting information

**S1 Checklist. The SPIRIT 2013 checklist.** Checklist outlining the trial design and planning according to SPIRIT 2013 guidelines.
(DOC)

**S2 Checklist. CONSORT 2010 checklist.** Checklist of information included in reporting the randomized trial per CONSORT 2010 standards.
(DOC)

## Acknowledgments

We sincerely thank Tiphaine Villaume for her invaluable help in reviewing, correcting, and providing insightful advice on the drafting of the Ethics Committee submission.

## Author contributions

**Conceptualization:** Marc Leone, Pierre-Hugues Roche, Frédéric Brocard.

**Funding acquisition:** Frédéric Brocard.

**Investigation:** Guillaume Baucher, Sylvie Liabeuf, Cécile Brocard, Lucas Troude, Marc Leone, Pierre-Hugues Roche, Frédéric Brocard.

**Methodology:** Guillaume Baucher, Sylvie Liabeuf, Cécile Brocard, Aurélie Ponz, Karine Baumstarck, Marc Leone, Pierre-Hugues Roche, Frédéric Brocard.

**Project administration:** Aurélie Ponz.

**Supervision:** Marc Leone, Pierre-Hugues Roche, Frédéric Brocard.

**Validation:** Aurélie Ponz, Karine Baumstarck, Marc Leone, Pierre-Hugues Roche, Frédéric Brocard.

**Visualization:** Frédéric Brocard.

**Writing – original draft:** Guillaume Baucher, Frédéric Brocard.

**Writing – review & editing:** Karine Baumstarck, Marc Leone, Pierre-Hugues Roche, Frédéric Brocard.

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
