## [Decision Letter · Decision Letter 0]

18 Mar 2025

PONE-D-25-05393The SpasT-SCI-T trial protocol: Investigating calpain-mediated sodium channel fragments as biomarkers for CNS injuries and spasticity predictionPLOS ONE

Dear Dr. Brocard,

Thank you for submitting your manuscript to PLOS ONE. After careful consideration, we feel that it has merit but does not fully meet PLOS ONE’s publication criteria as it currently stands. Therefore, we invite you to submit a revised version of the manuscript that addresses the points raised during the review process.

We look forward to receiving your revised manuscript.

Kind regards,

Mohammad Mofatteh, PhD, MPH, MSc, PGCert, BSc (Hons), MB BCh (c)

Academic Editor

PLOS ONE

Journal Requirements:

4. Please include a caption for figure 2.

5. Please remove all personal information, ensure that the data shared are in accordance with participant consent, and re-upload a fully anonymized data set.

Additional Editor Comments :

Also, please provide response to these comments:

1. The manuscript mentions that sodium channel fragments will be measured using a semi-automated Western blot, but does not describe antibody specificity or validation procedures. Provide details on antibody specificity, potential cross-reactivity, and assay validation steps to ensure reliability.

2. While emergency consent is addressed, the procedure for withdrawal of patients who regain capacity is not clearly outlined. Clarify how patients who regain capacity are re-consented and whether they can retrospectively withdraw data.

3. The study does not adequately account for confounding variables (e.g., age, sex, injury severity, concurrent medications).

4. Several minor grammatical errors and awkward phrasings should be revised for readability. Examples: "An upregulationg of persistent sodium currents..." → should be "an upregulation of persistent sodium currents..." "These insights may inform personalized therapeutic strategies for managing spasticity in SCI patients." → Consider rewording to "These insights could contribute to the development of personalized therapeutic strategies for spasticity management in SCI patients."

5. Figure 1 (CONSORT Diagram): Should be more detailed regarding patient enrollment and follow-up losses.

6. Spasticity and neurological recovery may continue beyond this period, limiting conclusions about long-term prognostic value. Consider an extended follow-up period (e.g., 12 months) or justify why 6 months is sufficient.

Reviewers' comments:

Reviewer's Responses to Questions

**Comments to the Author**

1. Does the manuscript provide a valid rationale for the proposed study, with clearly identified and justified research questions?

Reviewer #1: Partly

Reviewer #2: Yes

2. Is the protocol technically sound and planned in a manner that will lead to a meaningful outcome and allow testing the stated hypotheses?

Reviewer #1: Yes

Reviewer #2: Yes

3. Is the methodology feasible and described in sufficient detail to allow the work to be replicable?

Reviewer #1: Yes

Reviewer #2: Yes

4. Have the authors described where all data underlying the findings will be made available when the study is complete?

Reviewer #1: Yes

Reviewer #2: Yes

5. Is the manuscript presented in an intelligible fashion and written in standard English?

Reviewer #1: Yes

Reviewer #2: Yes

6. Review Comments to the Author

You may also provide optional suggestions and comments to authors that they might find helpful in planning their study.

Reviewer #1: This study can be considered of great importance in the area of central nervous system trauma management.

However, despite being well done and using excellent methodology, the justification for the study needs to be better targeted.

I only understood better that the focus was on traumatic injuries to the nervous system at the conclusion of the study.

I suggest that the authors consider improving this in the introduction of the article, improving the justification for why only individuals with TBI and SCI were included.

They should also consider including this issue focused on central nervous system trauma in the title of the study.

In addition, they should improve the discussion and organization of ideas, including in the discussion of the study.

I believe that, by making these adjustments, the article can be accepted.

Reviewer #2: Well written report on a very relevant topic.I am posistive this a very interesting papare for Plos One readers

7. PLOS authors have the option to publish the peer review history of their article (what does this mean? ). If published, this will include your full peer review and any attached files.

**Do you want your identity to be public for this peer review?** For information about this choice, including consent withdrawal, please see our Privacy Policy .

Reviewer #1: No

Reviewer #2: No

---

## [Author Response · Author response to Decision Letter 1]

8 Apr 2025

Dear Editor and Reviewers,

We sincerely thank you for your valuable feedback and the opportunity to revise our manuscript, "The SpasT-SCI-T trial protocol: Investigating calpain-mediated sodium channel fragments as biomarkers for central nervous system traumatic injuries and spasticity prediction." We appreciate the insightful comments, which have helped us improve the clarity and quality of our submission. Below, we address each comment point-by-point and detail the changes made to ensure compliance with PLOS ONE’s requirements and enhance the manuscript. All changes in the revised manuscript are highlighted in red for your convenience.

Journal Requirements:

RJ.1 Please ensure that your manuscript meets PLOS ONE's style requirements, including those for file naming.

We have ensured compliance with PLOS ONE’s style requirements. For file naming, all figures are cited sequentially in the text (Fig 1, Fig 2, Fig 3) and will be submitted as separate files named accordingly (Fig1.tif…) in an accepted format. Supporting information items are cited as S1 to S7 with specific categories (‘S1 Checklist,’ ‘S3 Consent Form’) and will be uploaded with matching file names (‘S1_Checklist.pdf,’ ‘S3_Consent_Form.pdf,’ ‘S4_Trial_Protocol.pdf’), using underscores as required. We have also corrected an inconsistent citation in the ‘Participants’ section, changing ‘S3 Supplemental Material’ to ‘S3 Consent Form’ to align with the file name and caption in the ‘Supporting Information’ section. These adjustments ensure full compliance with PLOS ONE’s style and file naming requirements.

RJ.2 Please include your full ethics statement in the ‘Methods’ section of your manuscript file. In your statement, please include the full name of the IRB or ethics committee who approved or waived your study, as well as whether or not you obtained informed written or verbal consent. If consent was waived for your study, please include this information in your statement as well.

We have added a full ethics statement to the “Methods and Analysis” section under a new subsection titled “Ethics Approval and Consent.” This statement includes the full name of the ethics committee that approved the study, the Comité de Protection des Personnes Est III, along with the approval details (CPP number 2024-A00265-42, dated October 7, 2024). It also clarifies that written informed consent was obtained from all participants or their legally authorized representatives, with specific procedures for emergency consent in cases of unconscious patients, as detailed in the “Participants” subsection and supported by the ethics approval letter (S6 Ethics Approval). Consent was not waived for this study. These revisions (pages 8-9; lines 190-202) ensure compliance with PLOS ONE’s requirements for a comprehensive ethics statement.

RJ.3 Please provide a complete Data Availability Statement in the submission form, ensuring you include all necessary access information or a reason for why you are unable to make your data freely accessible. If your research concerns only data provided within your submission, please write "All data are in the manuscript and/or supporting information files" as your Data Availability Statement.

We have updated the Data Availability Statement in the manuscript. Due to legal and ethical restrictions, the data, which include potentially identifying and sensitive participant information, cannot be made publicly available. They are accessible upon request from the sponsor, Assistance Publique des Hôpitaux de Marseille (Direction de la Recherche en Santé, 80 rue Brochier, Marseille, France, aap.drs@ap-hm.fr).

RJ.4 Please include a caption for figure 2.

We would like to clarify that a caption for Figure 2 is already included in the manuscript. It is inserted immediately following the paragraph where Figure 2 is first cited in the "Study Design" section of the "Methods and Analysis" part. If the referee has specific suggestions for additional details or modifications to the caption, we would be happy to incorporate them. Please let us know if further adjustments are required.

RJ.5 Please remove all personal information, ensure that the data shared are in accordance with participant consent, and re-upload a fully anonymized data set.

The manuscript under review is a clinical trial protocol for the SpasT-SCI-T study, which outlines the study design, methodology, and planned data collection without including any raw participant data or results. As such, it does not contain explicit personal information such as participant names, birth dates, or medical record numbers. All references to participants are generic and aggregated ("20 patients with spinal cord injury" or "10 healthy volunteers"), ensuring no identifiable data is presented. Regarding participant privacy and data anonymization, the protocol includes robust safeguards, as detailed in the "Data Management and Statistical Analysis" and "Legal and Ethical Aspects" sections.

We note that no data set is included with this protocol submission, as it describes a prospective study not yet conducted. To ensure full compliance with PLOS ONE policies, we reviewed all supplementary files listed in the CPP Est III opinion submitted to the journal. These files are templates or administrative documents and contain no participant-specific data.

RJ.6 Please review your reference list to ensure that it is complete and correct. If you have cited papers that have been retracted, please include the rationale for doing so in the manuscript text, or remove these references and replace them with relevant current references. Any changes to the reference list should be mentioned in the rebuttal letter that accompanies your revised manuscript. If you need to cite a retracted article, indicate the article’s retracted status in the References list and also include a citation and full reference for the retraction notice.

We carefully reviewed the entire reference list to ensure that all citations are accurate, complete, and up to date. We confirm that none of the cited articles have been retracted.

Additional Editor Comments:

RE.1 The manuscript mentions that sodium channel fragments will be measured using a semi-automated Western blot, but does not describe antibody specificity or validation procedures. Provide details on antibody specificity, potential cross-reactivity, and assay validation steps to ensure reliability.

We thank the reviewer for highlighting the need for clarity on antibody specificity and assay validation. In response, we have updated the ‘Biomarker Levels’ subsection in the ‘Methods and Analysis’ section to include detailed information. We use a monoclonal anti-pan Nav antibody (Sigma-Aldrich, CAT#S8809, 1:500) targeting the α-subunit of human sodium channels, specifically the conserved intracellular III-IV loop peptide “CTEEQKKYYNAMKKLGSKK.” Its specificity was validated by immunoblotting in HEK293 cells transfected with Nav channels, showing selective detection of the native channel (~250 kDa) and its cleavage fragments, with no signal in untransfected controls [21]. The conserved sequence across species reduces potential cross-reactivity. For assay validation, the semi-automated Western blot (Jess system, ProteinSimple) was optimized for sensitivity and reproducibility, with consistent detection of fragments across triplicate runs (coefficient of variation <10%) and normalization to total protein using the Jess Total Protein Detection Module to account for loading variability. These details enhance the reliability of our biomarker quantification and have been incorporated into the revised manuscript. These details are added on page 20, lines 470-476, within the ‘Biomarker Levels’ subsection.

RE.2 While emergency consent is addressed, the procedure for withdrawal of patients who regain capacity is not clearly outlined. Clarify how patients who regain capacity are re-consented and whether they can retrospectively withdraw data.

We thank the reviewer for this important observation. To address this, we have revised the “Participants” subsection in the “Methods and Analysis” section to clarify the re-consent process and withdrawal options for patients who regain capacity after emergency inclusion. When patients regain decision-making capacity, they are promptly approached by the investigator or a trained clinical team member with a detailed information sheet and verbal explanation of the study, including its objectives, procedures, risks, benefits, and their prior inclusion under emergency consent. They are then offered the opportunity to provide written continuation consent to remain in the study. If they decline, they may withdraw entirely, and we have now specified that they can request retrospective withdrawal of their data collected prior to regaining capacity, in compliance with French Public Health Code (Article L1122-1-1) and GDPR. Such data will be destroyed unless already anonymized and aggregated, ensuring ethical standards are upheld. These clarifications enhance transparency and participant autonomy in the protocol. These clarifications are added on page 13-14, lines 301-311and 323-325, within the ‘Participants’ subsection.

RE.3 The study does not adequately account for confounding variables (e.g., age, sex, injury severity, concurrent medications).

We thank the reviewer for raising this critical point about confounding variables. To address this, we have revised the “Statistical Methods” subsection in the “Methods and Analysis” section to clarify how we account for a comprehensive set of potential confounders, including but not limited to those highlighted (age, sex, injury severity, and concurrent medications). Our study collects extensive baseline and follow-up data (Table 3), encompassing age, sex, injury severity (ASIA grades A–D for SCI, GCS scores for TBI), and concurrent medications (e.g., antispasmodics, analgesics, antiepileptics), as well as additional variables such as comorbidities (e.g., diabetes, organ failure), trauma context (e.g., type of accident), and biological parameters (e.g., hemoglobin, renal function). In the updated statistical plan, we specify that multivariable linear regression models will adjust for these key confounders—specifically age, sex, injury severity, and concurrent medications as primary factors, with additional adjustments for comorbidities and other relevant variables from Table 3 as determined by their statistical significance and clinical relevance (e.g., prior studies [14, 21]). Subgroup analyses by injury severity are also refined to further explore its impact. This approach ensures that confounding effects on sodium channel fragment levels and outcomes like spasticity are rigorously controlled, enhancing the reliability of our findings. These updates are made on page 29, lines 664-671, within the ‘Statistical Methods’ subsection.

RE.4 Several minor grammatical errors and awkward phrasings should be revised for readability. Examples: "An upregulationg of persistent sodium currents..." → should be "an upregulation of persistent sodium currents..." "These insights may inform personalized therapeutic strategies for managing spasticity in SCI patients." → Consider rewording to "These insights could contribute to the development of personalized therapeutic strategies for spasticity management in SCI patients."

We have reviewed the entire manuscript to address other minor grammatical errors and awkward phrasings.

RE.5 Fig 1 (CONSORT Diagram): Should be more detailed regarding patient enrollment and follow-up losses.

We have updated Figure 1 (CONSORT flow diagram) to add details regarding patient enrollment and include a step after ‘Consent by proxy’ and ‘Emergency consent,’ showing that patients who regain capacity are re-consented, with options to continue or withdraw. The figure caption has also been revised to reflect these changes, enhancing clarity on the re-consent and withdrawal process. The figure has been updated and the revised caption is on page 8, lines 175-177.

RE.6 Spasticity and neurological recovery may continue beyond this period, limiting conclusions about long-term prognostic value. Consider an extended follow-up period (e.g., 12 months) or justify why 6 months is sufficient.

We thank the reviewer for this insightful comment. We have revised the ‘Study Design’ subsection in ‘Methods and Analysis’ to justify the 6-month follow-up period. Clinical signs of spasticity typically stabilize between 2 and 6 months post-injury, with electrophysiological measures showing minimal change thereafter (Hiersemenzel et al., 2000). Holtz et al. (2017) reported in a cohort of 465 SCI patients that the prevalence of problematic spasticity remained relatively stable, 35% at 1 year, 41% at 2 years, and 31% at 5 years, suggesting that 6 months captures the onset and severity of spasticity effectively. While neurological recovery (ASIA scale improvements) may continue beyond 6 months, our focus on early biomarker kinetics and spasticity prediction aligns with this timeline, as significant recovery plateaus within this period for many patients. Thus, a 6-month follow-up is sufficient for our primary objectives, though we acknowledge that extended follow-up (12 months) could be considered in future studies to assess longer-term outcomes. These clarifications, with references, are added to the manuscript. These clarifications are added on page 8, lines 165-168, within the ‘Study Design’ subsection.

1. Hiersemenzel L-P, Curt A, Dietz V. From spinal shock to spasticity: Neuronal adaptations to a spinal cord injury. Neurology 2000;54:1574–82.

2. Holtz KA, Lipson R, Noonan VK, et al. Prevalence and Effect of Problematic Spasticity After Traumatic Spinal Cord Injury. Archives of Physical Medicine and Rehabilitation 2017;98:1132–8.

Reviewer #1:

RR1.1 This study can be considered of great importance in the area of central nervous system trauma management. However, despite being well done and using excellent methodology, the justification for the study needs to be better targeted. I only understood better that the focus was on traumatic injuries to the nervous system at the conclusion of the study. I suggest that the authors consider improving this in the introduction of the article, improving the justification for why only individuals with TBI and SCI were included.

We thank the reviewer for this important remark. We recognize your point that the rationale for specifically targeting individuals with TBI and SCI could be more explicitly targeted in the introduction. While we cited preclinical studies (Brocard et al., 2016; Von Reyn et al., 2009) showing sodium channel cleavage in these traumatic conditions, we may not have sufficiently clarified why these two injury types were selected over other CNS conditions. To address this, we propose revising the introduction to emphasize that SCI and TBI were chosen due to their traumatic etiology, the high prevalence of motor disorders like spasticity in SCI (affecting ~75% of patients), and the shared molecular mechanism of calpain-mediated sodium channel cleavage observed in preclinical models of both conditions. This revision will also briefly note that non-traumatic CNS conditions (stroke or neurodegenerative diseases) were excluded because they lack comparable evidence of this specific mechanism in the context of our biomarker hypothesis. These revisions are made on page 5, lines 99–107, within the ‘Introduction’ section.

RR1.2 They should also consider including this issue focused on central nervous system trauma in the title of the study.

The current title, "The SpasT-SCI-T trial protocol: Investigating calpain-mediated sodium channel fragments as biomarkers for CNS injuries and spasticity prediction", uses "CNS injuries" to encompass traumatic SCI and TBI, as clarified in the introduction. To make this focus explicit, we propose revising it to: "The SpasT-SCI-T trial protocol: Investigating calpain-mediated sodium channel fragments as biomarkers for traumatic CNS injuries and spasticity prediction". Adding "traumatic" aligns with your recommendation and sharpens the scope.

RR1.3 In addition, they should improve the discussion and organization of ideas, including in the discussion of

---

## [Editor Report · Decision Letter 1]

17 Apr 2025

The SpasT-SCI-T trial protocol: Investigating calpain-mediated sodium channel fragments as biomarkers for traumatic CNS injuries and spasticity prediction

PONE-D-25-05393R1

Dear Dr. Brocard,

We’re pleased to inform you that your manuscript has been judged scientifically suitable for publication and will be formally accepted for publication once it meets all outstanding technical requirements.

Kind regards,

Dr. Mohammad Mofatteh, PhD, MPH, MSc, PGCert, BSc (Hons), MB BCh (c)

Academic Editor

PLOS ONE

Additional Editor Comments (optional):

The authors have addressed the reviewers comments adequately. The manuscript has been improved.
---

## [Editor Report · Acceptance letter]

PONE-D-25-05393R1

PLOS ONE

Dear Dr. Brocard,

I'm pleased to inform you that your manuscript has been deemed suitable for publication in PLOS ONE. Congratulations! Your manuscript is now being handed over to our production team.

Kind regards,

on behalf of

Dr. Mohammad Mofatteh

Academic Editor

PLOS ONE